# The 12-Membered TNFR1 Peptide, as Well as the 16-Membered and 6-Membered TNF Peptides, Regulate TNFR1-Dependent Cytotoxic Activity of TNF

**DOI:** 10.3390/ijms25073900

**Published:** 2024-03-31

**Authors:** Daria M. Yurkina, Elena A. Romanova, Anna V. Tvorogova, Zlata K. Naydenysheva, Alexey V. Feoktistov, Denis V. Yashin, Lidia P. Sashchenko

**Affiliations:** 1Institute of Gene Biology (RAS), Moscow 119334, Russia; yrkina121@gmail.com (D.M.Y.); elrom4@rambler.ru (E.A.R.); nzkchemistry@mail.ru (Z.K.N.); sashchenko@genebiology.ru (L.P.S.); 2Center for Precision Genome Editing and Genetic Technologies for Biomedicine, Institute of Gene Biology, Russian Academy of Sciences, Moscow 119334, Russia; annatvor@mail.ru; 3Engelhardt Institute of Molecular Biology (RAS), Moscow 119334, Russia

**Keywords:** PGLYRP1, TNFR1, cytotoxicity, tumor cells, apoptosis, necroptosis, short peptides

## Abstract

Understanding the exact mechanisms of the activation of proinflammatory immune response receptors is very important for the targeted regulation of their functioning. In this work, we were able to identify the sites of the molecules in the proinflammatory cytokine TNF (tumor necrosis factor) and its TNFR1 (tumor necrosis factor receptor 1), which are necessary for the two-stage cytotoxic signal transduction required for tumor cell killing. A 12-membered TNFR1 peptide was identified and synthesized, interacting with the ligands of this receptor protein’s TNF and Tag7 and blocking their binding to the receptor. Two TNF cytokine peptides interacting with different sites of TNFR1 receptors were identified and synthesized. It has been demonstrated that the long 16-membered TNF peptide interferes with the binding of TNFR1 ligands to this receptor, and the short 6-membered peptide interacts with the receptor site necessary for the transmission of a cytotoxic signal into the cell after the ligands’ interaction with the binding site. This study may help in the development of therapeutic approaches to regulate the activity of the cytokine TNF.

## 1. Introduction

The study of the interaction between cytokines and their analogues with their specific receptors contributes to the understanding of the mechanisms of the immune response and to the development of new approaches in immunotherapy [1].

The TNF cytokine is one of the most studied regulatory proteins. It was discovered as a tumor necrosis factor and named due to its ability to induce the death of tumor cells [2,3]. TNF is a pro-inflammatory cytokine. It plays an important role in the regulation of the immune response through the induction of a pro-inflammatory response to pathological infections [4,5]. The main source of soluble TNF is monocytic cells, but they are known for being secreted by lymphocytes [6]. TNF, in the acute phase of infection, affects intercellular contacts and adhesion molecules, making it easier for cells of the immune system to access the site of infection from the circulatory system. As can be seen from the above, TNF plays an important role in the antimicrobial protection of the body and a systemic decrease in its secretion level, for example, with the help of anti-TNF antibodies, implies significant side effects, such as poor control of tuberculosis infection [7]. In addition to its active role in the fight against infection, TNF is known as a key immunoregulator in the development of adaptive immunity. Its importance for immune system development is evidenced by the fact that, although the TNF family now has more than 20 members homologous to it and with partially overlapping functions, TNF knockout is lethal [8]. It is precisely due to its crucial role in the body and the many different functions that it performs, interacting with only two specific cellular receptors 1 and 2, that it is an important task to specifically target only one of these functions. Further research has expanded the understanding of its biological activity. Currently, TNF is known to be a classic pro-inflammatory cytokine that also plays an important role in the development of inflammatory autoimmune diseases such as rheumatoid arthritis, sepsis and inflammation-associated proliferation of tumor cells [9,10,11,12,13]. Thus, TNF can perform opposite actions in the body. On the one hand, it induces programmed cell death in tumor cells, and protects against the development of oncological diseases. On the other hand, it promotes the proliferation of tumor cells. It should also be borne in mind that TNF plays a key role in the development of a pro-inflammatory immune response and enhances the expression of cell adhesion molecules [14,15].

The mechanisms of TNF-dependent signal transduction are well known. Two TNF receptors have been described: TNFR1 and TNFR2 [16,17,18]. TNFR1 is expressed on almost all cell types and is involved in the pro-inflammatory immune response and the development of inflammation [19]. TNFR2 is present mainly on cells of the immune system and performs mainly an immunomodulatory role.

When TNF interacts with TNFR1, the intracellular part of TNFR1 binds with the adaptor protein TRADD. This event leads to the assembly of an intracellular complex, which includes caspases and serine protein kinases RIP1 and RIP3 [20,21]. Depending on the activation of these enzymes, alternative processes of cell death are induced in the cells: apoptosis and necroptosis [22,23]. The activation of ubiquitinase blocks the processes of cell death and leads to the activation of the transcription factor NFkß [24]. Under the control of NFkß, the immune response, cell growth, tissue differentiation and gene expression of proinflammatory cytokines are regulated [25,26,27]. TNF overexpression leads to the increased expression of inflammatory cytokine genes, which ultimately causes TNF-dependent inflammation (Figure 1). To prevent the development of inflammatory diseases, compounds that reduce the amount of soluble TNF or prevent signal transduction via the TNF–TNFR1 system are needed.

To downregulate the destructive activity of the TNF cytokine pathway, several approaches are used. One of them is the use of antibodies to either TNF of TNFR1 proteins. This approach is used in the treatment of several diseases, for example, rheumatoid arthritis, and has several advantages, such as high specificity and a sufficient lifetime of the active molecules in the blood of the patients, but also several drawbacks, such as high costs and autoimmune problems [28]. Another approach is the identification of active peptide fragments of regulatory proteins that model the functional action of full-sized proteins. Such peptides, by binding to the receptor, can inhibit or activate its function. Treatment of TNF–TNFR1 pathway activation using small peptides also has its advantages, such as good penetration of the internal barriers of the patient, low immune response and potentially low cost, but also drawbacks such as the lower lifetime of the active molecules in the patient’s blood.

The aim of this study was to search for peptide fragments of TNF and TNFR1 that cause the cytotoxic activity of this cytokine. It was necessary to identify the following: (1) the TNFR1 peptide fragment interacting with its ligands; (2) a TNF peptide fragment inhibiting the cytotoxic activity of TNFR1 ligands; (3) TNF peptides simulating the cytotoxic activity of this TNFR1 ligand.

## 2. Results

### 2.1. The Structure of the TNF Protein Complex with Its Receptor

Figure 2 shows the crystal structure of the TNF complex with its TNFR2. The regions of TNF and TNFR1 that interact during the formation of the receptor–ligand complex are labeled. It can be seen that the single TNF monomer interacts with two TNFR2 monomers. A comparative analysis of the amino acid sequence allowed us to establish a 58% identity between the TNFR2 fragment (NWVPECLSCGSRCS) interacting with TNF and the corresponding TNFR1 fragment (NHLRHCLSCSKC). The 16-membered (NPQAEGQLQWLNRRAN) TNF_L_ and 6-membered (YVLLTH) TNF_S_ TNF fragments interacting with the receptor, as well as the 12-membered peptide fragment TNFR1 (NHLRHCLSCSKC), were synthesized, and the effect of these peptides on the transmission of intracellular signals induced by the TNFR1 receptor was investigated.

### 2.2. The 12-Membered TNFR1 Peptide Interacts with Ligands of This Receptor and Inhibits Their Cytotoxic Activity

First, the affinity of the interaction of the TNFR1 peptide with its TNF and Tag7 ligands was evaluated using microscale thermophoresis [29,30]. Figure 3a shows the binding of the TNF protein with the TNFR1 peptide. Figure 3b displays signals from binding Tag7 with the TNFR1 peptide. Both graphs show clear changes in thermophoretic signals depending on the concentration of the TNFR1 peptide. As a control, the interaction of the fluorescent-labeled TNF with a full-size exodomain of the TNFR1 receptor (sTNFR1) was studied (Figure 3c). The calculated K_d_ indicates a high affinity among the studied complexes (Table 1).

The K_d_ values for the full-size TNF–peptide TNFR1 complex, full-size Tag7–peptide TNFR1 complex and full-size TNF–sTNFR1 complex are nanomolar, which suggests the stability of these complexes. It can be seen that the values of K_d_ differ marginally, which indicates the comparable affinity of these complexes.

Previously, we showed that the K_d_ value for the Tag7–sTNFR1 complex is 42.2 nm. The K_d_ value for the Tag7–TNFR1 peptide complex is approximately 10 times lower. It can be assumed that Tag7 has a high affinity with the sTNFR1 peptide fragment.

Thus, we have demonstrated that the 12-membered TNFR1 peptide can interact with TNF and with Tag7 to form fairly stable complexes.

Next, we investigated how this interaction affects the cytotoxic activity of TNFR1 ligands. TNF and the Tag7–Hsp70 complex were preincubated with the 12-membered peptide of TNFR1, and the cytotoxic effect of these complexes was evaluated. It is known that TNF binding to the TNFR1 receptor could induce cell death via different mechanisms with different kinetics. Cell apoptosis is a fast process of several hours. During necroptosis, cell death processes take place after a few dozen hours. Taking into account this ability of TNFR1 to participate in the induction of alternative cytotoxic pathways that develop at different time intervals, cytotoxic activity was determined after 3 h and after 20 h (Figure 4). It can be seen that both TNF and the Tag7–Hsp70 complex completely lose both apoptotic and necroptotic activity when interacting with the 12-membered peptide of TNFR1. It can be assumed that the inhibition of the cytotoxic effect of the ligands is associated with the competition of a 12-membered peptide with the cell surface receptor on for binding to the ligands. Interestingly, the synthesized TNFR1 fragment interacts not only with its specific TNF ligand, but also with the Tag7–Hsp70 complex.

### 2.3. The 16-Membered TNF_L_ Peptide Binds to the TNFR1 Receptor and Inhibits the Cytotoxic Effect of Its Ligands

The use of microscale thermophoresis allowed us to characterize the affinity of the interaction of the synthesized 16-membered TNF_L_ peptide and sTNFR1 (Figure 5a). A clear binding curve representing the dependence of the thermodynamic signal on the concentration of TNF_L_ peptide allowed us to calculate K_d_. A low K_d_ value (9.1 ± 3.5 nM) indicates a high affinity among the interacting reagents and suggests the ability of the TNF_L_ peptide to form a stable complex with sTNFR1.

Next, the interaction of the synthesized TNF_L_ peptide with the cells was studied using confocal microscopy (Figure 6). The presented results demonstrate that TNFR1 is present on cells (green) and that the TNF_L_ peptide binds to cells (red). One can see the colocalization of the peptide with the receptor (yellow), which indicates their interaction. Statistical calculations have shown that about 95% of the cells contain a receptor and that 69,7% of the cells contain sites of the colocalization of the peptide with a receptor. A total of 100 fields were calculated, the typical two fields of which are shown in Appendix A.

The study of the effect of the interaction of the TNF_L_ peptide with TNFR1 on the cytotoxic effect of its ligands revealed the inhibitory effect of this peptide. The preincubation of cells with TNF_L_ led to the disappearance of cytotoxic activity developing under the action of both TNF and the Tag7–Hsp70 complex (Figure 7). Both apoptosis (3 h) and necroptosis (20 h) were inhibited. Consequently, the peptide blocks access to the TNFR1 epitope responsible for binding to ligands, and it complicates the interaction of the cytotoxicity-inducing ligands with this receptor fragment.

### 2.4. The 6-Membered TNF_S_ Peptide Interacts with the TNFR1 Receptor, but Does Not Inhibit the Cytotoxic Effect of Its Ligands

Using microscale thermophoresis, it was found that the synthesized TNF_S_ peptide, as well as the synthesized TNF_L_ peptide, have a high affinity with the sTNFR1 exodomain. The K_d_ value was 1.4 ± 0.5 nM (Figure 5b). It can be assumed that these peptides can form a stable complex with TNFR1. The interaction of the TNF_S_ peptide with a receptor on the cell membrane was shown using confocal microscopy. The results shown in Figure 8 demonstrate the colocalization of the peptide with the receptor, indicating their binding.

Statistical analysis shows 93% of the cells carrying the TNFR1 receptor on the outer membrane and 71% of the cells stained with the TNF_S_ peptide. A total of 100 fields were calculated, two typical fields of which are shown in Appendix A.

Figure 5 shows the results of the TNF_S_ peptide effect on the cytotoxic effect of the ligands. It can be seen that, unlike TNF_L_, TNF_S_ does not inhibit apoptosis nor necroptosis which develop under the action of TNF or the Tag7–Hsp70 complex.

Therefore, although TNF_S_ has a high affinity for the TNFR1 receptor and forms a stable complex on the cell membrane, it does not prevent TNFR1 ligands from binding to the receptor and activating intracellular signals.

### 2.5. TNF_L_ and TNF_S_ Peptides Simulate the Cytotoxic Effect of TNF

Next, we verified the cytotoxic activity of TNF peptides (Figure 9). It can be seen that no single peptide has cytotoxic activity. However, the simultaneous addition of TNF_L_ and TNF_S_ peptides to the cells completely reproduces the cytotoxicity of full-size TNF (Figure 9a).

Blocking the cytotoxic activity of the TNF_L_ and TNF_S_ peptides’ combination by using antibodies to TNFR1 allows us to suggest that cell death is induced through the interaction of peptides with the TNFR1 receptor. Also, as it was shown, alternative processes of cell death are induced via the TNFR1 receptor [31] in cells under the action of peptides, namely caspase-dependent apoptosis and RIP1-dependent necroptosis (Figure 9b,c). Previously, we showed that the TNFR1 receptor ligand Tag7–Hsp70 complex induces a necroptotic signal involving lysosomes and mitochondria [32]. Here, we also investigated the involvement of these organelles in the development of necroptosis under the action of the sum of TNF_L_ and TNF_S_ peptides. To this end, cytotoxic activity was determined in the presence of inhibitors blocking the permeabilization of lysosomal membranes, enzymatic activity of lysosomal enzymes and accumulation of reactive oxygen species (ROS). It can be seen that inhibitors of the Ca^2+^-dependent protease of calpain, cathepsins B and D, as well as the addition of the antioxidant ionyl, lead to the disappearance of cytotoxic activity (Figure 9d). It can be assumed that lysosomes and mitochondria are involved in the induction of necroptosis under the action of the combination of TNF peptides.

## 3. Discussion

This paper expands on the understanding of the mechanisms of interaction between the cytotoxic factor TNF and its TNFR1. The peptide epitopes of the TNFR1 receptor and its TNF ligand providing the functional activity of these full-sized proteins have been characterized.

As mentioned above, TNF plays a key role in the active phase of inflammation. The interaction of TNF with its TNFR1 induces the appearance of a wide range of cytokines and chemokines involved in various pathological processes, as well as in immune protection. To regulate these processes, it is advisable to act upon both components of the TNF–TNFR1 receptor complex, causing a decrease in TNF concentration or blocking ligand binding to the receptor and subsequent intracellular signal transmission.

It has been established that the TNF–TNFR1 system causes opposite effects (for example, death or proliferation of tumor cells); therefore, to regulate its action, it is desirable to selectively suppress its activity.

A promising approach may be to identify epitopes of ligand and receptor molecules that perform separate effects and obtain peptide fragments that perform various functions.

Based on this approach, a TNF peptide (aa. 150–170), capable of killing tumor cells and inhibiting tumor growth, has been identified [33].

Recently, we have identified peptide fragments of the TNFR1 ligand, the Tag7 protein, which performs only one function: either the inhibition of the TNF interaction with the receptor, or the cytotoxic effect on cells in complexes with the Hsp70 protein [34]. It has been shown that anti-TNF antibodies that reduce TNF levels result in a decrease in the development of rheumatoid arthritis and sepsis [28].

Here, we searched for TNF and TNFR1 receptor peptides that can cause both a decrease in ligand concentration and the blocking of its receptor.

The 12-membered TNFR1 peptide and two peptides of its specific TNF ligand were synthesized: the 16-membered TNF_L_ peptide and the 6-membered TNF_S_ peptide located at the sites of receptor–ligand interaction (Appendix A), and the functional activity of these peptides during cytotoxic signal transmission was investigated.

The 12-membered TNFR1 peptide showed high affinity with two TNFR1 ligands with different amino acid sequences. The K_d_ values for the TNF cytokine and innate immunity protein Tag7 were comparable. It can be assumed that this peptide determines the specificity of the interaction of the receptor with its ligands. The ability of the peptide to bind to ligands to form a stable complex causes competition between this peptide and TNFR1 on the cell membrane for interaction with the ligand and leads to the inhibition of cytotoxic activity induced by TNFR1 ligands.

The crystal structure of the TNF–TNFR2 complex suggests the binding of the TNF monomer to different peptide sites of two different TNFR1 monomers.

The TNF_L_ and TNF_S_ peptides have a high affinity with the soluble exodomain of the TNFR1 receptor and can bind to this receptor on the cell membrane. However, they have different functional activities. As well as the 12-membered TNFR1 peptide, TNF_L_ inhibits the cytotoxic action of TNFR1 ligands. However, the mechanism of inhibition by these two peptides is different. The TNF_L_ peptide interacts with a receptor on the cell membrane, possibly with a 12-membered peptide site of the receptor, and it prevents the binding of other ligands to this receptor. TNF_L_ itself has no cytotoxic effect. TNF_S_ also has no cytotoxic effect and does not inhibit the cytotoxic activity of other ligands. It seemingly binds to another site of the receptor responsible for activating the receptor after the ligand is fixed on the binding site. Consequently, two regions of the TNFR1 exodomain are involved in the activation of the intracellular signal.

The Tag7 we studied earlier, as well as the TNF_L_ peptide, inhibited the cytotoxic activity of ligands, but did not have cytotoxicity. The amino acid sequence of Tag7 seemingly lacks a peptide fragment interacting with the second site of the receptor. For cytotoxic action, Tag7 requires the presence of a coactivator protein capable of binding to this site. Only the Tag7-Hsp70 complex induced cell death, as in the case of TNF_L_ and TNF_S_. When studying the mechanisms of programmed cell death under the action of the combination of TNF_L_ and TNF_S_ peptides, it was found that peptides, as well as full-size TNF, induce alternative cytotoxic processes: apoptosis and necroptosis. Apoptosis occurs with the participation of specific proteases caspase-8 and caspase-3. During the induction of necroptosis, the specific RIP1 phosphokinase is activated. The participation of Ca^2+^-dependent protease calpain, lysosomal enzymes of cathepsins and reactive oxygen species (ROS) in the induction of necroptosis has also been shown. The Ca^2+^-dependent protease calpain is involved in the destabilization of the lysosomal membrane and promotes the appearance of lysosomal enzymes in the cytosol [35]. Cathepsins play a significant role in programmed cell death, as they are able to exhibit proteolytic activity at pH values in the cytoplasm of cells. It is known that mitochondria can be the target of lysosomal enzymes [36]. It has been shown that cathepsins B and D are capable of causing the permeabilization of mitochondrial membranes. The destabilization of mitochondrial membranes can lead to a decrease in membrane potential and to the accumulation of ROS. The inhibition of cell death by the action of the combination of TNF_L_ and TNF_S_ in the presence of calpain and cathepsin inhibitors, as well as an antioxidant, indicates the participation of lysosomes and mitochondria in this process.

Enhancing the activity and suppressing the toxicity of TNF is a serious problem in modern immunology. Modifications of the TNF molecule are used to solve it. It has been shown that the cleavage of seven amino acids from the N-terminus or point mutations of the TNF molecule leads to an increase in antitumor activity and a decrease in lethal toxicity in mice [37]. Another promising approach to solving this problem was to obtain TNF peptide fragments that simulate the activity of a full-size molecule. It has been shown that an evolutionarily conservative peptide derived from TNF is capable of causing apoptosis or necrosis in cancer cells. This P16 peptide also inhibits tumor growth and metastasis [33]. The 6-aminoacids TNFR1 peptide binds to TNF, causing the inhibition of the p38 signaling pathway and inflammation [38]. A fusion molecule RMP16 containing a specific peptide from human serum albumin, a peptide for cleavage by Factor Xa and a 20-membered bioactive peptide TNF P16 was obtained using genetic engineering technology. RMP16 effectively inhibits the proliferation of DU145 prostate cancer cells through caspase-dependent apoptosis and arrest of the C6/C1 cell cycle [39].

Previously, using limited trypsinolysis, we identified the peptide 17.1-protein Tag7, which causes the bifunctional activity of this TNFR1 ligand: inhibition of the cytotoxic action of TNF or induction of cytotoxic activity using the Hsp70 coactivator. Shortened fragments of this peptide were also identified, responsible for only one function: inhibitory or cytotoxic [34]. Inhibitory peptides had a noticeable protective effect on the development of CFA-dependent autoimmune arthritis [40]. 

Here, we have described three new peptide fragments participating in TNF–TNFR1 interaction. Two peptides, TNFR1 and TNF_L_, had inhibitory activity. Two peptides, TNF_L_ peptide and TNF_s_, modeled the cytotoxic activity of full-sized TNF, causing apoptosis or necroptosis in cancer cells.

Thus, TNF peptides have been identified as exhibiting functional activity at different stages of interaction with TNFR1. TNF_L_ ensures the interaction of this ligand with the ligand-binding site of the receptor; TNF_S_ activates the intracellular signal after the binding of the ligand on the surface of the receptor. The resulting peptides can be used in the creation of drugs for anti-inflammatory and antitumor therapy.

## 4. Materials and Methods

### 4.1. Cell Cultivation and Sorting

L929 cells were cultured in DMEM with 2 mm L-glutamine and 10% FCS (Invitrogen, Carlsbad, CA, USA). This cell line was obtained from the cell line collection of the N. N. Blokhin National Medical Research Center of Oncology of the Ministry of Health of Russia.

### 4.2. Proteins and Antibodies

To obtain the soluble component of the human TNFR1 receptor, the following primers were used: TNFR1(sol)-for: 5′-GCATATGAGTGTGTGTGTCCCCAAGGAAA.

TNFR1(sol)-rev: 5′-GCTCGAGATTCTCAATCTGGGGTAGGC.

The cDNA fragment was cloned into the pET22b plasmid and expressed in a bacterial system (strain BL21) induced by 1 mM IPTG at 37 °C for 5 h. Proteins were purified on Ni-NTA Agarose (Thermo Fisher Scientific, Waltham, MA, USA) according to the manufacturer’s protocol.

cDNAs of recombinant human Tag7, Hsp70 and human TNF-α (GenBank access numbers: NM_005091, NM_005345 and NM_000594, respectively) were subcloned in pQE-31 plasmid and expressed in E.coli M15 (pREP4) (Qiagen, Hilden, Germany). Hsp70 and TNF-α were purified with Ni-NTA Agarose (Thermo Fisher Scientific, Waltham, MA, USA) in accordance with the manufacturer’s instructions.

TNF_L_, TNF_S_ and TNFR1 peptides were synthesized as described in [31].

For cytotoxicity tests, target cells were cultured in 96-well plates (3 × 10^4^ cells per well), and cytotoxic complexes were added (1 nM). Cytotoxicity was measured after 20 h of incubation by Cytotox 96 analysis kit (Promega, Madison, WI, USA) in accordance with the manufacturer’s protocol. In inhibition assays, cells were pre-incubated for 1 h with the Caspase 3 inhibitor Ac-DEVD-CHO (5 μM) (Thermo Fisher Scientific, Waltham, MA, USA), caspase 8 inhibitor Ac-IEID-CHO (5 μM) (Thermo Fisher Scientific, Waltham, MA, USA), RIP1 kinase inhibitor necrostatin 1 (5 μM) (Thermo Fisher Scientific, Waltham, MA, USA), ionol (1 µM), cathepsin B inhibitor Ca-074Me (10 µM), cathepsin D inhibitor pepstatin A (10 µM) and peptide inhibitor of calpain (10 µM) (all from Sigma-Aldrich, St. Louis, MO, USA). All these agents were added 1 h before incubation of the cells with the TNF, Tag7–Hsp70 complex or combination of TNF_L_ and TNF_S_ peptides. Antibodies to TNF (Catalog. No. A11534) were obtained from ABclonal (ABclonal, Woburn, MA, USA). Antibodies to TNFR1 were obtained from Santa Cruz Biotechnology, Dallas, TX, USA).

### 4.3. Microscale Thermophoresis

A fluorescent label was added to the purified sTNFR1 protein using the Alexa Fluor™ 633 Protein Labeling Kit (Life Technologies Corporation, Eugene, OR, USA) in accordance with the manufacturer’s protocol. A fluorescent label was added to the purified Tag7 and TNF proteins using the Alexa Fluor™ 488 Protein Labeling Kit (Life Technologies Corporation, Eugene, OR, USA) in accordance with the manufacturer’s protocol. TNF_L_, TNF_S_ and sTNFR1 (C = 200 nM) were incubated for 30 min in the dark at room temperature in 16 different concentrations obtained by sequential dilution, starting with the highest soluble concentration. The samples were transferred to glass capillaries (Monolith NT Capillaries) and analyzed by microscale thermophoresis on a Nano-Temperature Monolith NT 115 device (30% IR laser power). The signal quality was monitored using a NanoTemper Monolith device to detect possible autofluorescence of the ligand, aggregation or changes in the rate of photobleaching. The experiments were carried out in at least three repetitions and processed using affinity analysis software (MO Control v.1.6.1, NanoTemper Technologies GmbH, München, Germany).

### 4.4. Confocal Microscopy

L929 cells were grown on glass coverslips and fixed with a cold solution of 4% formaldehyde. Cells were rinsed 3 times with PBS, and then the samples were placed into blocking solution (1% BSA in PBS) for 20 min at room temperature. The TNFR1 receptor was stained with sheep antibodies against TNFR1 and Goat anti-sheep IgG (H + L) Cross-Adsorbed Secondary Antibody, Alexa Fluor™ 633 (Molecular Probes By Life Technologies, Carlsbad, CA, USA). The TNF_L_ (TNF_S_) peptides were stained with polyclonal rabbit antibodies against TNF-α and Goat anti-Rabbit IgG (H + L) Cross-Adsorbed Secondary Antibody, Alexa Fluor™ 488. After washing with PBS, the coverslips were embedded in ProLong Gold (Thermo Fisher Scientific, Waltham, MA, USA). Fluorescence images were obtained using a Leica STELLARIS 5 confocal microscope (Leica, Wetzlar, Germany), analyzed using Leica confocal software (2.61.15) and processed in Photoshop CE (Adobe Systems, San Jose, CA, USA).

### 4.5. Statistical Analysis

The data were analyzed using Statistica 6.1 (StatSoft^®^) software. The Shapiro–Wilk test was used to confirm the normality of the distribution of the data. The results are presented as an average value ± SD. Statistically significant differences were determined using the *t*-test. A *p* < 0.05 value was considered statistically significant. GraphPad Prism 6 software was used for data presentation.

## 5. Conclusions

In this study, we were able to identify the sites of the molecules of the proinflammatory cytokine TNF and its TNFR1, which are necessary for the two-stage cytotoxic signal transduction required for tumor cell killing. We synthesized the 12-membered peptide of TNFR1, able to bind ligands of TNFR1, and two peptides of TNF, binding to two different sites on TNFR1. One of these peptides interferes with ligand binding, while the other one is required for cytotoxic signal transduction. In this work, we were able to identify two sites of the TNF–TNFR1 complex, which may be used for the modulation of the TNF cytotoxic signal transduction. It was discovered in this work that peptides may be used in antitumor therapy or in anti-cytokine treatment.

## Figures and Tables

**Figure 1 ijms-25-03900-f001:**
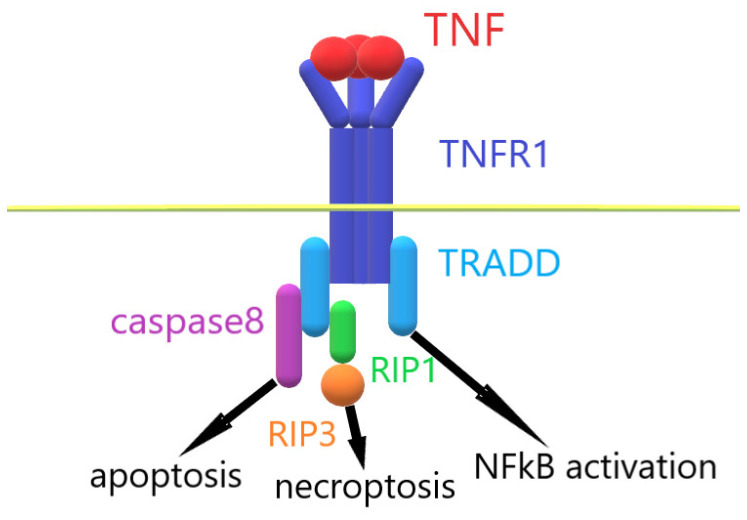
Mechanism of TNF–TNFR1 interaction.

**Figure 2 ijms-25-03900-f002:**
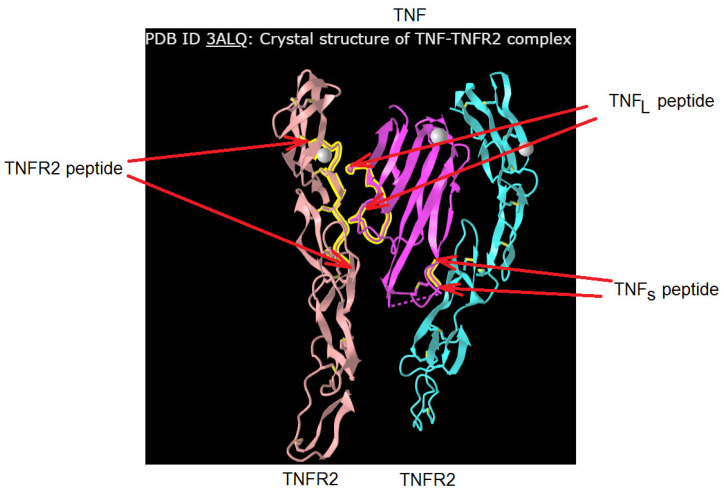
Designation of the peptides of TNFR2 (in this work, its analogue from TNFR1 structure was used) and TNF. The original structure was found in https://www.ncbi.nlm.nih.gov/ PDB ID: 3ALQ, accessed on 1 January 2024.

**Figure 3 ijms-25-03900-f003:**
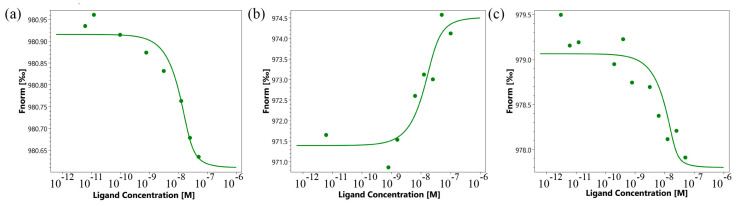
Binding of TNF with TNFR1 peptide (**a**), Tag7 with TNFR1 peptide (**b**) and TNF with sTNFR1 (**c**) using microscale thermophoresis. Binding of all compounds to labeled proteins resulted in a clear response in fluorescence signal, dependent on the concentration of the compound. Graphs are represented as Fnorm [‰] against ligand concentration. Data represent three independent experiments and were fitted to a Kd binding model assuming a 1:1 binding stoichiometry.

**Figure 4 ijms-25-03900-f004:**
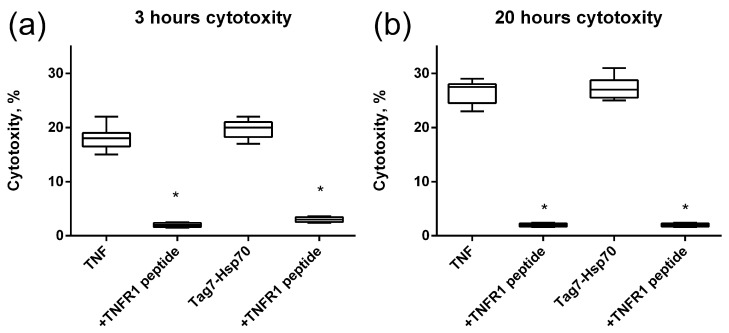
Cytotoxic activity of the TNF and Tag7–Hsp70 complex (1 nM) on L929 cells after 3 (**a**) and 20 (**b**) hours in the presence of TNFR1 peptide (10nM) is shown. *n* = 5 for each point, (* *p*-value < 0.05).

**Figure 5 ijms-25-03900-f005:**
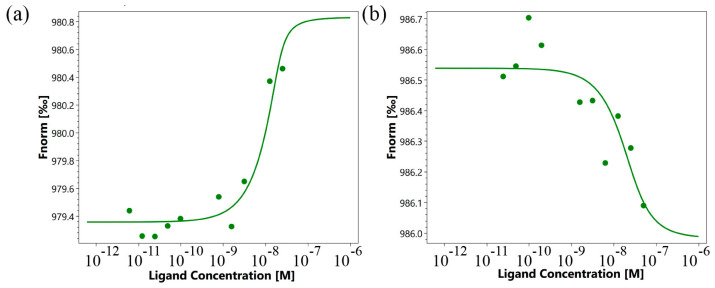
Binding of TNF_L_ (**a**) and TNF_S_ (**b**) peptides with sTNFR1 using microscale thermophoresis. Binding of all compounds to labeled sTNFR1 protein resulted in a clear response in fluorescence signal, dependent on the concentration of the compound. Graphs are represented as Fnorm [‰] against ligand concentration. Data represent three independent experiments and were fitted to a Kd binding model assuming a 1:1 binding stoichiometry.

**Figure 6 ijms-25-03900-f006:**
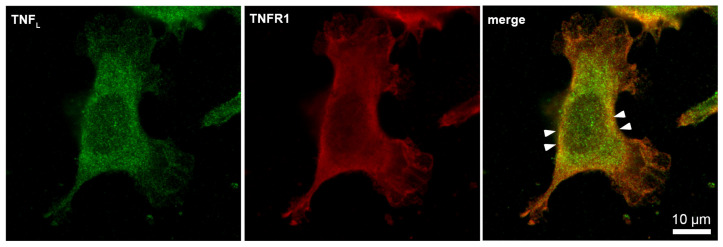
Confocal micrograph of TNF_L_ peptide (green) and TNFR1 (red) and layers superposition on the surface of L929 cell. Arrows indicate colocalization region. About 70% of all observed cells were double-stained (*n* = 4).

**Figure 7 ijms-25-03900-f007:**
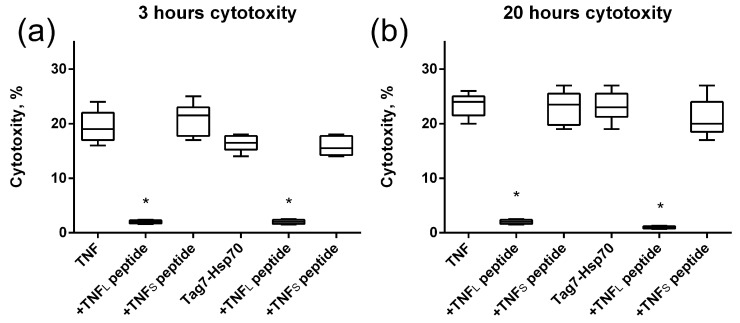
Cytotoxic activity of TNF and the Tag7–Hsp70 complex (1 nM) on L929 cells after 3 (**a**) and 20 (**b**) hours in the presence of TNF_L_ and TNF_S_ peptides (10 nM) is shown. *n* = 5 for each point, (* *p*-value < 0.05).

**Figure 8 ijms-25-03900-f008:**
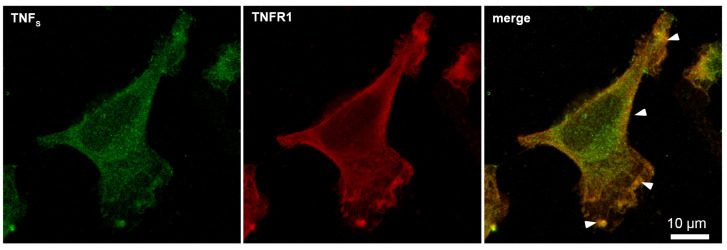
Confocal micrograph of TNF_S_ peptide (green) and TNFR1 (red) and layers’ superposition on the surface of L929 cell. Arrows indicate colocalization region. About 71% of all observed cells were double-stained (*n* = 4).

**Figure 9 ijms-25-03900-f009:**
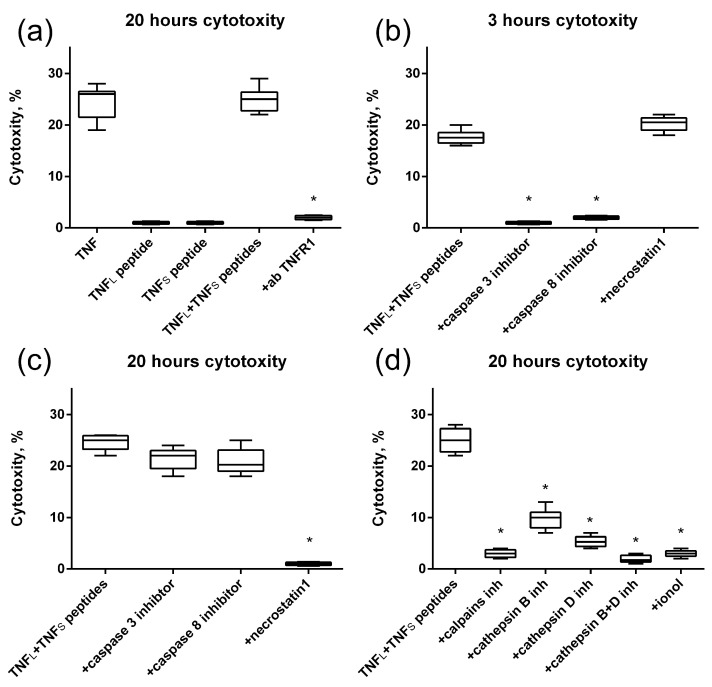
Cytotoxic activity of the TNF, TNF_L_ peptide, TNF_S_ peptide and simultaneous addition of both TNF_L_ and TNF_S_ peptides on L929 cells after 20 h of incubation. The TNFR1 antibodies were added to the sum of TNF_L_ and TNF_S_ peptides prior to incubation (**a**). Cytotoxic activity of the sum of TNF_L_ and TNF_S_ peptides on L929 cells after 3 (**b**) and 20 (**c**) hours in the presence of caspase 3 and 8 inhibitors (5 μM) and necrostatin1 (5 μM) is shown. (**d**) Cytotoxic activity of the sum of TNF_L_ and TNF_s_ peptides on L929 cells after 20 h of incubation in the presence of calpains inhibitors (10 μM), cathepsins inhibitors (10 μM) and antioxidant (1 μM). *n* = 5 for each point, (* *p*-value < 0.05).

**Table 1 ijms-25-03900-t001:** Dissociation constants obtained via microscale thermophoresis.

Ligands	K_d_, nM
TNF–TNFR1 peptide	2.4 ± 0.5
Tag7–TNFR1 peptide	4.5 ± 1.2
TNF–sTNFR1	1.2 ± 0.2

## Data Availability

Data are contained within the article and Appendix A.

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
