# Peer review of "The 12-Membered TNFR1 Peptide, as Well as the 16-Membered and 6-Membered TNF Peptides, Regulate TNFR1-Dependent Cytotoxic Activity of TNF"

_ijms, 2024, doi:10.3390/ijms25073900_

Round 1

Reviewer 1 Report

Comments and Suggestions for Authors

Manuscript ID: ijms-2916028; Title: The 12-membered TNFR1 peptide, as well as the 16-membered and 6-membered TNF peptides regulate TNFR1-dependent cytotoxic activity of TNF. In this paper, the author's focus was on identifying the essential sites of the proinflammatory cytokine TNF and its TNFR1 receptor required for the two-stage cytotoxic signal transduction crucial for tumor cell killing. They synthesized a 12-membered peptide targeting TNFR1 receptor, capable of binding TNFR1 ligands, along with two peptides of TNF that bind to distinct sites on TNFR1 receptor. The authors claim that their findings can potentially be utilized in antitumor therapy.

The topic is interesting for the referee. Considering the importance of the current hot topic antitumor therapy, the reviewer suggests that this manuscript can be accepted after the following minor issues are addressed.

1) The author needs to write the full forms of TNF (tumor necrosis factor), TNFR1 (tumor necrosis factor receptor 1), etc., at the beginning of the paper for better understanding.

2) It would be important for the author to include a graphical image to show the mechanism described on page 2. Graphical aids can greatly enhance understanding, especially when explaining complex processes. So, need to include graphical images.

3)  It is important for the author to improve the introduction by providing a thorough review of the state-of-the-art, including an introduction to the advantages and disadvantages of the research. Additionally, including the motivation and necessity of the research will help provide context and clarity for readers.

4) It would be beneficial to move Supplemental Figure 1 from the supplementary information (SI) to the main paper for better visibility and understanding.

5) It is important for all figures (like Figure 1, Figure 2, and Figure 5...) to have the same scale on both axes for standard representation and comparison

6) In the discussion section, it would be convincing for the author to include a comparison of the present work with previous studies to highlight advancements or differences. Additionally, explaining the importance of the current work can help readers understand its significance in the field.

Comments on the Quality of English Language

Minor editing of the English language required

Author Response

The topic is interesting for the referee. Considering the importance of the current hot topic antitumor therapy, the reviewer suggests that this manuscript can be accepted after the following minor issues are addressed.

  • The author needs to write the full forms of TNF (tumor necrosis factor), TNFR1 (tumor necrosis factor receptor 1), etc., at the beginning of the paper for better understanding.

We thank the reviewer for careful reading of our work and useful comments. We have changed the Abstract Section in response to reviewer’s comments.

  • It would be important for the author to include a graphical image to show the mechanism described on page 2. Graphical aids can greatly enhance understanding, especially when explaining complex processes. So, need to include graphical images.

We have added new scheme in the Introduction Section to address this issue.

  • It is important for the author to improve the introduction by providing a thorough review of the state-of-the-art, including an introduction to the advantages and disadvantages of the research. Additionally, including the motivation and necessity of the research will help provide context and clarity for readers.

We have modified the Introduction Section to address the reviewers comments.

  • It would be beneficial to move Supplemental Figure 1 from the supplementary information (SI) to the main paper for better visibility and understanding.

We have moved the Supplemental Figure 1 from Supplemental Materials to the Results Section.

  • It is important for all figures (like Figure 1, Figure 2, and Figure 5...) to have the same scale on both axes for standard representation and comparison.

We have changed the scale of new Figures 4,7,9 to make them clearer for the potential readers.

6) In the discussion section, it would be convincing for the author to include a comparison of the present work with previous studies to highlight advancements or differences. Additionally, explaining the importance of the current work can help readers understand its significance in the field.

We have modified the Discussion Section to address this comment.

Comments on the Quality of English Language

Minor editing of the English language required

We have done the language polishing to improve the manuscript.

Reviewer 2 Report

Comments and Suggestions for Authors

The study is innovative and the proposed hypothesis is justified. The authors, identified the sites of the proinflammatory cytokine TNF and its receptor TNFR1 molecules that are required for the two-step cytotoxic signal transduction necessary to kill tumor cells. Identification and synthesis of a novel 12-membered peptide of TNFR1 may help develop therapeutic approaches to regulate TNF activity. The matherials and methods part are given in details for replicate the proposed experimental procedures and analysis.The results and discussion parts are presented in details. Minor editing of English language required.

Remarks:

1. ........"Cell apoptosis is fast process that generally takes several hours to disrupt the cells outer membrane. Cell necroptosis takes a longer time interval to show the death phenotype" to be reworked.  Row 47-49; 63-65; 81-83; to be reworked;

2. The results prezented to Figure 2, 5, 7 will be better to be presnt as violins; or on SPSS grafs;  the deviation value to be add ;

3. The description of the figures not enough.

4. the conclussion part does not respond in detail to the innovativeness of the research- to be reworked.

5. The references part - all used references are appropriate and from the last 3-5 years; but references are not prepared according to the requirements of the journal - bolt/ not bold.

Comments on the Quality of English Language

Minor editing of English language required.

Author Response

Remarks:

  1. ........"Cell apoptosis is fast process that generally takes several hours to disrupt the cells outer membrane. Cell necroptosis takes a longer time interval to show the death phenotype" to be reworked.  Row 47-49; 63-65; 81-83; to be reworked;

We thank the reviewer for careful reading of our work and useful comments. We have corrected this issue.

  1. The results prezented to Figure 2, 5, 7 will be better to be presnt as violins; or on SPSS grafs;  the deviation value to be add ;

We have changed the presentation of our data in new Figures 4,7,9 to SPSS graphical standarts.

  1. The description of the figures not enough.

We have changed the figures legends to address this issue.

  1. the conclussion part does not respond in detail to the innovativeness of the research- to be reworked.

We have reworked the Conclusion Section in response to the reviewers comments.

  1. The references part - all used references are appropriate and from the last 3-5 years; but references are not prepared according to the requirements of the journal - bolt/ not bold.

We thank the reviewer for careful reading of our text. We have reworked the references by the Journal standards.

Comments on the Quality of English Language

Minor editing of English language required.

We have done some language polishing to improve the text of the manuscript.

Round 2

Reviewer 2 Report

Comments and Suggestions for Authors

-

Author Response

We thank the reviewer for useful comments.